# The Comparative Safety of Epirubicin and Cyclophosphamide versus Docetaxel and Cyclophosphamide in Lymph Node-Negative, HR-Positive, HER2-Negative Breast Cancer (ELEGANT): A Randomized Trial

**DOI:** 10.3390/cancers14133221

**Published:** 2022-06-30

**Authors:** Deyue Liu, Jiayi Wu, Caijin Lin, Shuning Ding, Shuangshuang Lu, Yan Fang, Jiahui Huang, Jin Hong, Weiqi Gao, Siji Zhu, Xiaosong Chen, Ou Huang, Jianrong He, Weiguo Chen, Yafen Li, Kunwei Shen, Li Zhu

**Affiliations:** 1Department of General Surgery, Comprehensive Breast Health Center, Ruijin Hospital, School of Medicine, Shanghai Jiaotong University, Shanghai 200025, China; 15026629865@163.com (D.L.); pinkscorpio@163.com (J.W.); cjlin20@fudan.edu.cn (C.L.); nannanyard@163.com (S.D.); trista_lu@126.com (S.L.); fangyan4743@163.com (Y.F.); hjh3269@163.com (J.H.); pingdimuhj@163.com (J.H.); gaoweiqi2733@163.com (W.G.); zsj_mu@yeah.net (S.Z.); chenxiaosong0156@hotmail.com (X.C.); ou_huang@126.com (O.H.); hejrong@hotmail.com (J.H.); cwg-dr@hotmail.com (W.C.); 13601792038@139.com (Y.L.); kwshen@medmail.com.cn (K.S.); 2Department of Breast and Thyroid Surgery, Shanghai General Hospital, School of Medicine, Shanghai JiaoTong University, No. 100 Haining Road, Hongkou District, Shanghai 200080, China

**Keywords:** epirubicin, docetaxel, neutropenia, adverse events, breast cancer

## Abstract

**Simple Summary:**

Anthracycline and taxane-based chemotherapy are the cornerstone of adjuvant therapy for early breast cancer. In recent years, several trials explored the efficacy of the anthracycline-free regimens, especially for HER2-negative breast cancer patients, which turned out to be a feasible alternative. However, there is no comparison between epirubicin and cyclophosphamide versus docetaxel and cyclophosphamide about their safety and efficacy to date. ELEGANT is the first phase III randomized trial comparing the safety and efficacy of EC versus TC, and it reports comprehensive safety profiles of both regimens with a primary endpoint of grade 3 to 4 neutropenia rate.

**Abstract:**

Background: In adjuvant settings, epirubicin and cyclophosphamide (EC) and docetaxel and cyclophosphamide (TC) are both optional chemotherapy regimens for lymph node-negative, hormone receptor (HR)-positive, human epidermal receptor 2 (HER2)-negative breast cancer patients. Neutropenia is one of the most common adverse events (AEs) of these regimens. The rate of grade 3–4 neutropenia varies in different studies, and direct comparisons of safety profiles between EC and TC are lacking. Method: ELEGANT (NCT02549677) is a prospective, randomized, open-label, noninferior hematological safety trial. Eligible patients with lymph node-negative HR+/HER2-tumors (1:1) were randomly assigned to received four cycles of EC (90/600 mg/m^2^) or TC (75/600 mg/m^2^) every three weeks as adjuvant chemotherapy. The primary endpoint was the incidence of grade 3 or 4 neutropenia defined by National Cancer Institute-Common Terminology Criteria for Adverse Events (NCI-CTCAE) version 4.0 on an intention-to-treat basis. Noninferiority was defined as an upper 95% CI less than a noninferiority margin of 15%. Results: In the intention-to-treat population, 140 and 135 patients were randomized into the EC and TC arms, respectively. For the primary endpoint, the rate of grade 3 or 4 neutropenia is 50.71% (95% CI: 42.18%, 59.21%) in the EC arm and 48.15% (95% CI: 39.53%, 56.87%) in the TC arm (95%CI risk difference: −0.100, 0.151), showing the noninferiority of the EC arm. For secondary endpoints, the rate of all-grade anemia is higher in the EC arm (EC 42.86% versus TC 22.96%, *p* = 0.0007), and more patients suffer from nausea/vomiting, hair loss, and nail changes (*p* < 0.01) in the EC arm. No statistically different disease-free survival was observed between the two arms (*p* = 0.13). Conclusion: EC is not inferior to TC in the rate of grade 3 or 4 neutropenia, but more other AEs were observed in the EC group.

## 1. Introduction

Adjuvant chemotherapy is an essential part of the comprehensive treatment of breast cancer. Chemotherapy reduces 10-year breast cancer mortality by one third according to EBCTCG results. [1]. One of the common adverse events (AE) of myelosuppressive chemotherapy is neutropenia, which is closely related to a higher risk of infection, longer days of hospitalization and higher cost [2]. Neutropenia has become the most common reason to blame for dose reduction and chemotherapy delays [3], which has increased death risk compared with patients without chemotherapy modifications in a retrospective analysis (HR 2.76, 95%CI 1.3–5.7, *p* < 0.05) [4]. Granulocyte colony-stimulating factors (G-CSF) is an effective treatment for neutropenia but was reported to have a correlation with higher incidences of acute myeloid leukemia and myelodysplastic syndrome [5,6]. Therefore, safety as well as efficacy should both be considered when making medical decisions.

According to NCCN guidelines, docetaxel and cyclophosphamide (TC) and anthracycline and cyclophosphamide (AC/EC) are all optional chemotherapy regimens for lymph node-negative, hormone receptor (HR)-positive, human epidermal growth receptor 2 (HER2)-negative breast cancer patients [7]. In the aspect of safety profile, the incidence rate of grade 3–4 neutropenia was reported to reach 61% in the TC (75/600 mg/m^2^) arm and 55% in the AC (60/600 mg/m^2^) arm in the US Oncology Research Trial 9735 [8]. Earlier studies showed that the hematological equitoxic dose ratio of doxorubicin (DXR) to epirubicin (EPI) is about 1:1.2 [9]. As a result, a standard EC regimen (90/600 mg/m^2^) theoretically would cause higher hematological toxicities than AC (60/600 mg/m^2^). In clinical trials, EC demonstrated grade 3–4 neutropenia rates of 8.4–54.2% [10,11,12], and TC showed 41–61% [8,13,14]. Based on the data above, the assumption could be made that EC (90/600 mg/m^2^) might be noninferior to TC (75/600 mg/m^2^) in terms of neutropenia rate. With regard to efficacy, four cycles of TC (75/600 mg/m^2^) have been proven to be superior to standard AC (60/600 mg/m^2^) in both disease-free survival (DFS) and overall survival (OS) [15].

To date, no direct comparisons of safety and efficacy have been made between TC and EC, so this study is aimed at comparing the safety profiles of EC versus TC to make a better chemotherapy choice for node-negative, HR-positive, HER2-negative breast cancer patients.

## 2. Methods

### 2.1. Trial Design

EC vs. TC in Lymph node-negative, ER-positive, HER2-negative Breast Cancer as Adjuvant Chemotherapy (ELEGANT) is a prospective, randomized, open-label, noninferior trial (ClinicalTrial.gov identifier: NCT02549677, registered on 15 September 2015) conducted at the Comprehensive Breast Health Center, Ruijin Hospital, Shanghai Jiao Tong University School of Medicine (RJBC). Patients were informed of the study purpose, procedure of the trial, potential adverse events, estimated expenses for chemotherapy, and obligations before enrollment. The first subject was enrolled on 8 September 2015. A total of 294 patients were evaluated for the safety profile and adverse events of EC versus TC regiments in HR+/HER2-breast cancer patients as adjuvant therapy.

This trial was carried out in accordance with the Declaration of Helsinki and Good Clinical Practice Guidelines, and all subjects provided written informed consent. The protocol, consent and relative documents were approved by independent ethics committee in Ruijin Hospital (Number 2015 [55]).

Adult female patients younger than 70 years old diagnosed as invasive breast cancer and operated in RJBC with a life expectancy of more than 12 months were included in the study. The surgical specimens were examined in the Pathological Department of Ruijin Hospital to ensure all participants had ER- or PR-positive, HER2-negative, and lymph node-negative tumors. ER- or PR-positive was defined as nuclear-stained cells accounting for more than 1% of tumor cells. HER2-negative was defined as 0–1+ by immunohistochemical (IHC) analysis or negative by fluorescence in situ hybridization (FISH) test. The adjuvant therapy scheme was approved by a multidisciplinary team at RJBC. Adjuvant radiotherapy and endocrine therapy regimen were administered when needed. Baseline blood routine tests and other basic tests were also administered to make sure all participants were in normal hematopoietic, liver and renal function. We excluded patients who were allergic, intolerant or poorly compliant with the regimen; previously treated or metastatic breast cancer patients; patients previously treated with anthracycline or taxane or combined with other malignant tumors (except for controlled cervical carcinoma in situ or skin basal cell carcinoma). Patients who had ≥1 grade of peripheral neuropathy; who were pregnant or lactating; or who were previously or concurrently enrolled in another trial were also excluded.

The random number list was 1:1 generated by computer to allocate patients into the EC (experimental) arm or the TC (controlled) arm. After randomization, no blinding was performed. EC (epirubicin 90 mg/m^2^, cyclophosphamide 600 mg/m^2^) and TC (docetaxel 75 mg/m^2^, cyclophosphamide 600 mg/m^2^) were both given intravenous q3w for four cycles.

### 2.2. Assessment

The primary end point was the incidence of grade 3 or 4 neutropenia (defined as serum neutrophil granulocyte level <1.0 × 10^9^/L and ≥0.5 × 10^9^/L for grade 3; <0.5 × 10^9^/L for grade 4). Primary prophylactic granulocyte colony-stimulating factor was not allowed. The secondary end points were other severe hematological toxicities, 3-year disease-free survival (DFS) and 3-year overall survival (OS). Apart from hematological AEs, any chemo-related AEs and therapies were also recorded by investigators. The adverse events were assessed throughout the whole treatment period until three weeks after the last course and graded according to the National Cancer Institute-Common Terminology Criteria for Adverse Events (NCI-CTCAE) version 4.0. Details are presented in Appendix A. Participants received routine blood testing at least twice a week and hepatorenal function tests every three weeks.

Safety data were regularly reviewed by investigators. Patients were also followed up for physical examination every 3 months; for hepatorenal function test, tumor markers and breast and abdominal ultrasound examination every 6 months; and for chest CT scan and mammography every year after surgery.

### 2.3. Statistical Analysis

The null hypothesis of the trial was that the rate of grade 3 or 4 neutropenia of experimental group is not inferior to that in the control group. To detect noninferiority, we allowed a difference of up to 15% in the primary outcome. Assuming a neutropenia rate of 40% in EC arm, we needed an enrollment of 152 patients per arm for a two-sided test to rule out the prespecified difference in 95% confidence interval (CI) of the noninferiority, allowing for a 10% patient dropout rate at a two-sided significance level of 0.05 with 80% power. To conclude noninferiority, the upper bound of the 95% CI should be more than the prespecified margin of 0.15. If noninferiority was established, the upper limit of the 95% CI could be further compared with 0 for assessment of superiority.

The safety population was composed of patients who received at least one cycle of chemotherapy and had recorded safety profiles. Safety analyses were carried out in the safety population. Univariate and multivariate logistic analyses were performed to identify possible predictors of grade 3 or 4 neutropenia. The full analysis set (FAS) comprised the per-protocol population and the drop-out population. The intention-to-treat (ITT) population included patients receiving at least one cycle of chemotherapy without severe protocol violation. Efficacy end point analyses were performed in the ITT population with Kaplan–Meier method. The statistical analyses were carried out with R software (version 3.6.1; http://www.R-project.org, accessed on 10 May 2020) and SPSS software version 23 (IBM Corporation, Armonk, NY, USA). Two-sided *p* value < 0.05 was considered statistically significant.

## 3. Results

### 3.1. Patients and Tumor Characteristics

From August 2015 to March 2020, 294 participants were recruited and randomized to receive either EC (*n* = 147) or TC (*n* = 147). Fourteen objects refused or lost to follow-up during their chemotherapy courses. Three patients received primary prophylaxis of neutropenia, which was not allowed in the schema. Two patients deviated from the protocol. Finally, 275 patients were included in the safety population and 292 in the intention-to treat population.

The patients’ baseline characteristics were well balanced between arms (Table 1). The median age was 51 in the whole safety population. Most patients (90.55%) had normal neutrophil count according to the reference value in Ruijin Hospital. Around three quarters, 206 patients (74.91%), had normal weight, and 58 (21.09%) patients were overweight or obese classified by body mass index (BMI). Most patients (150, 54.55%) had at least one comorbidity. Regarding treatment, 58 (41.43%) patients received G-CSF in the EC group and 40 (29.63%) in the TC group (*p* = 0.055).

### 3.2. Primary Endpoint

In terms of the primary endpoint, 71 (50.71%, 95% CI: 42.18%, 59.21%) in EC and 65 (48.15%, 95% CI: 39.53%, 56.87%) in TC group experienced at least one cycle of grade 3 or higher neutropenia (95% CI risk difference: −0.100, 0.151) (Table 2). The upper bounds of 95% CI for the risk difference between the two groups was more than the predefined margin of 0.15. The study met its primary endpoint of indicating EC regimen was noninferior to TC in the aspect of grade 3 or 4 neutropenia rate.

### 3.3. Secondary Endpoints

In total, 131 (93.57%, 95% CI 87.79%, 96.83%) patients in EC suffered all-grade neutropenia and 100 (74.07%, 95% CI 65.69%, 81.05%) in TC did so (*p* < 0.001) (Table 2). The mean first occurrence cycle of grade 3 or higher neutropenia was earlier in EC (1.46 vs. 1.89, *p* = 0.009). The rates of grade 3–4 neutropenia in the two groups during four cycles are shown in Figure 1. Among all the cycles, the rates of grade 3 or 4 neutropenia after the first cycle ranked the highest in both groups (30.22% in total, 38.41% in the EC arm and 21.54% in the TC arm). Compared with patients without grade 3 or 4 neutropenia in the first cycle, patients with grade 3 or 4 neutropenia in the first cycle experienced a higher rate of grade 3 or 4 neutropenia in the following cycles (51.85% vs. 27.81%, *p* = 0.0155).

More all-grade anemia was observed in EC group (42.86% vs. 22.96%, *p* < 0.001). Both groups experienced similar rates of all grade thrombopenia (7.14% vs. 4.44%, *p* = 0.49) and all grade hepatotoxicity (11.43% vs. 8.15%, *p* = 0.48). Grade 3 or higher anemia (1 vs. 0), thrombopenia (2 vs. 1) and hepatotoxicity (1 vs. 1) are rare in both groups (Table 3).

Non-hematological AEs (NHAE) are shown in Figure 2 and Table 4. The most common NHAEs (all grade) were nausea/vomiting, hair loss, nail change, fatigue and ostalgia. Most NHAEs were generally similar between the two arms except that nausea/vomiting, hair loss and nail change were more common in the EC group (87.86% vs. 55.56%, *p* < 0.001, 90.00% vs. 76.30%, *p* < 0.01, 54.29% vs. 22.22%, *p* < 0.001 separately).

Treatment discontinuation rates were similar between groups (1 [0.71%] in EC vs. 1 [0.74%] in TC), as were dose modification (0 vs. 1 [0.74%]) and treatment delay (8 [5.71%] vs. 4 [2.96%]). The AE that mostly led to treatment delay was hepatotoxicity (3 [2.14%] vs. 3 [2.22%]).

### 3.4. Predictors of Grade 3 or 4 Hematological AEs

Taking all the patient characteristics into univariate analysis, none of the factors was associated with grade 3 or 4 neutropenia. With a threshold of 0.1, factors including comorbidity, BMI and surgery were taken into multivariate logistic analysis. However, no independent factors were recognized (Table 5).

The rates of grade 3 or 4 anemia (1 in EC group and 0 in TC group) and thrombocytopenia (2 in EC group and 1 in TC group) were quite low in the safety population; thus predictors for them were not calculated.

### 3.5. Disease-Free Survival in Both Groups

In the ITT population (292 patients, 147 in EC and 145 in TC), 286 patients were disease-free at median follow-up of 33 months, while 1 in EC and 5 patients in TC had relapse or distant metastasis. There was no statistical difference observed in disease-free survival (DFS) between the two groups (HR 4.51 (95%CI 0.52, 38.80), log-rank *p* = 0.13, Figure 3).

## 4. Discussion

The current study compared the safety profile of EC versus TC with a primary endpoint of grade 3–4 neutropenia. Our analysis showed that the rate of grade 3 or 4 neutropenia in EC was noninferior to that in TC in patients with node-negative, HR-positive tumors. Most adverse events were comparable between the two groups, but more non-hematological adverse events like nausea, vomiting, hair loss and nail change were reported in EC.

Anthracycline- and taxane-based regimens are the backbone cytotoxic treatment for early breast cancer [1], but to date, there is no consensus about the optimal drug combination, doses and courses. Taxane is widely used in the early breast cancer setting, with or without the combination with anthracycline. The 7-year follow-up of the US Oncology Research (USOR) Trial 9735 demonstrated that four cycles of TC was superior to AC with a tolerable toxicity [15]. Consequently, the National Comprehensive Cancer Network (NCCN) Guidelines version 4.2022 recommend TC (75/600 mg/m^2^) for four cycles as one of the preferred regimens for adjuvant chemotherapy for early breast cancer [7]. On the other hand, EPI showed similar efficacy to that of doxorubicin (DXR), and a higher dose of EPI in EC regimen (90/600 mg/m^2^) might exhibit higher efficacy than AC (60/600 mg/m^2^) [16]. To date, no trial has compared EC with TC head to head in terms of either efficacy or safety. To our knowledge, ELEGANT is the first prospective, randomized, open-label clinical trial that compared the safety profiles of EC versus TC.

Previously reported grade 3 and 4 neutropenia rates for EC and TC demonstrated a quite wide range. In a feasibility study of TC for 6 cycles in Japan, the rate of grade 3 and 4 neutropenia was 41% with the permission of G-CSF use when grade 4 neutropenia or febrile neutropenia developed [13]. The USOR 9735 trial reported TC with rates of 61% in 2006 and 58% in 2009 [8,15]. For EC, in a German study of 6 cycles of EC, a rate of 73% was reported [17], while an English study of 4 cycles of EC reported a rate of 16% [12]. A higher dose of EPI (120 mg/m^2^) combined with cyclophosphamide for 4 cycles comes with a higher rate of severe neutropenia, 81.6% in a Italian study in 2005 [18] and 54.2% in another Italian study in 2012 [11]. Evidence suggested that there is great variability in pharmacokinetics, pharmacodynamics and tolerance of antitumor drugs between different ethnicities [19]. Several studies also showed that Asian patients exhibited a high rate of hematological AEs than Caucasian patients [20,21,22]. In our study, EC and TC demonstrated a rate of 50.71% and 48.15% respectively, providing evidence for toxicity and tolerance in Chinese cohorts.

In case of non-hematological AEs, nausea and vomiting are the leading non-hematological AEs in the current study (87.86% in EC group and 55.56% in TC group, *p* < 0.001), more frequent than previous studies [15,23], but grade 3–4 was rare in both groups. Cardiotoxicity is a remarkable non-hematological AE of EPI, but no cardiac impairments were observed in either group.

Apart from the safety profiles of both regimens, possible predictors of adverse events were also analyzed in our study. Previous studies suggested that obese patients experienced fewer hematological toxicities than lean women. A feasibility study of TC in Japan showed that normal or underweight women (BMI < 25) were 2.6 times more likely to experience grade 3 or higher hematological toxicities than obese women (BMI ≥ 30) [24]. A retrospective study in French found age >60 and BMI ≤ 30 as risk factors for anemia in anthracycline-based regimens with or without taxanes [25]. In the present study, however, neither age nor BMI was a risk factor for grade 3 or 4 neutropenia with either regimen. This might have arisen from the low proportions of obese women in both groups (22.14% in EC and 20.00% in TC). The use of G-CSF in both arms (41.43% in EC and 29.63% in TC, *p* = 0.055) might also confound the risk of potential factors [26].

A standard chemotherapy protocol for HER2-negative early breast cancer (EBC) is in suspense despite many relevant clinical trials. An anthracycline-free regimen has been highly discussed in recent years. ABC trials demonstrated a higher efficacy of TaxAC versus six cycles of TC in high-risk HER2-negative EBC, but the superiority did not persist in an HR-positive subgroup [23]. Meanwhile, another trial of similar regimen came to a negative result [27]. A meta-analysis of four randomized trials comparing TC and A+T regimens showed that sequential A+T was associated with a higher risk of toxicity but no clear survival benefit [28]. In the current study focusing particularly on low-risk luminal breast cancer patients, an anthracycline-free regimen might be an optimal choice given a full consideration of efficacy and toxicity. Our study is the first to report efficacy results for EC vs. TC, providing evidence for the low-risk luminal breast cancer setting.

There are some limitations of the study. Firstly, ELEGANT was an open-label trial. Apart from the primary endpoint, some other adverse events were subjectively reported by unmasked patients, which might bring about biases. Secondly, the median follow-up time is 33 months. No death was observed during the follow-up period, and only 6 recurrences occurred. Longer follow-up is needed for the detection of recurrence or death in patients with HR-positive, HER2-negative and node-negative breast cancer [29].

## 5. Conclusions:

EC is noninferior to TC in grade 3 or 4 neutropenia rates. EC exhibited more non-hematological AEs such as nausea/vomit, hair loss and nail change than TC. There was no difference in terms of DFS between the two arms. TC is a preferred adjuvant regimen for patients with node-negative, HR-positive, HER2-negative early breast cancer, while EC is an appropriate alternative to TC with tolerable toxicities.

## Figures and Tables

**Figure 1 cancers-14-03221-f001:**
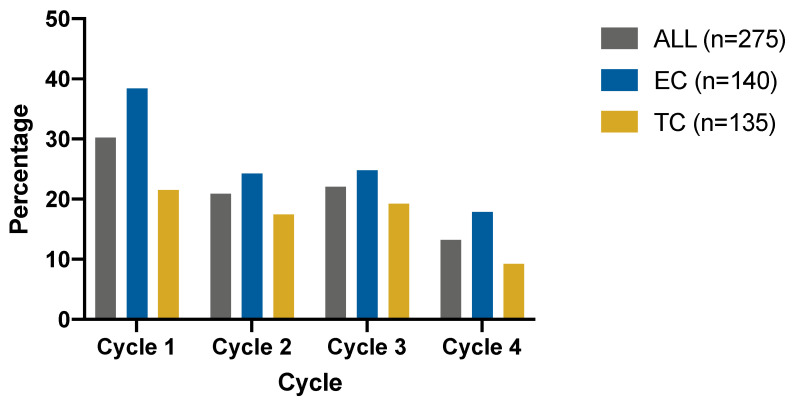
The rate of grade 3 or 4 neutropenia by different cycles and different regimens. Abbreviations: EC = epirubicin and cyclophosphamide; TC = docetaxel and cyclophosphamide.

**Figure 2 cancers-14-03221-f002:**
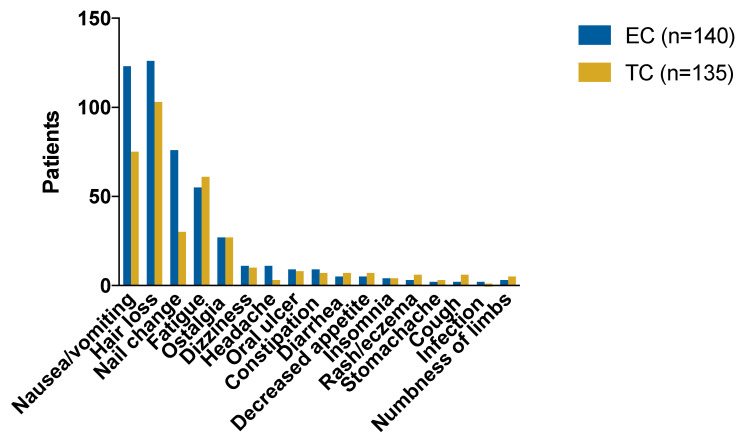
The number of common non-hematological all-grade adverse events by regimen. More nausea/vomiting (87.86% vs. 55.56%, *p* < 0.001), hair loss (90.00% vs. 76.30%, *p* < 0.01) and nail change (54.29% vs. 22.22%, *p* < 0.001) were reported in EC. Abbreviations: EC = epirubicin and cyclophosphamide; TC = docetaxel and cyclophosphamide.

**Figure 3 cancers-14-03221-f003:**
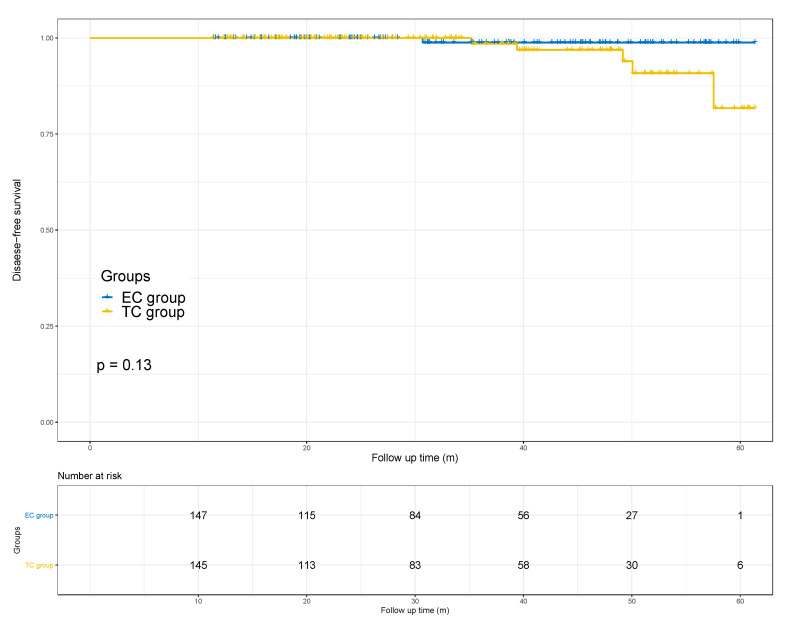
The comparison of the disease-free survival rates between the two groups. With a median follow-up of 33 months, no significant difference was observed between two groups (HR 4.51 [95% CI 0.52, 38.80], log-rank *p* = 0.13). Abbreviations: EC = epirubicin and cyclophosphamide; TC = docetaxel and cyclophosphamide.

**Table 1 cancers-14-03221-t001:** The patient and treatment characteristics by arm (safety population).

	EC *n* = 140 (%)	TC *n* = 135 (%)	*p*
Age at diagnosis			0.554
<60	108 (77.14)	99 (73.33)	
≥60	32 (22.86)	36 (26.67)	
Body mass index, kg/m^2^			0.463
Underweight (<18.5)	4 (2.86)	7 (5.19)	
Normal weight (18.5–24.9)	105 (75.00)	101 (74.81)	
Overweight (25–29.9)	27 (19.29)	20 (14.81)	
Obese (≥30)	4 (2.86)	7 (5.19)	
Number of comorbidities			0.974
0	63 (45.00)	62 (45.93)	
≥1	77 (55.00)	73 (54.07)	
G-CSF			0.055
Yes	58 (41.43)	40 (29.63)	
No	82 (58.57)	95 (70.37)	
Surgery			0.107
Mastectomy	82 (58.57)	65 (48.15)	
Breast conserving	58 (41.43)	70 (51.85)	
T Stage			0.320
1	105 (75.00)	93 (68.89)	
2	35 (25.00)	42 (31.11)	
PR status			0.219
Negative	25 (17.86)	16 (11.85)	
Positive	115 (82.14)	119 (88.15)	
Ki-67			0.242
<14%	43 (30.71)	32 (23.70)	
≥14%	97 (69.29)	103 (76.30)	
LVI			0.226
No	130 (92.86)	121 (89.63)	
Yes	6 (4.29)	12 (8.89)	
Unknown	4 (2.86)	2 (1.48)	
Grade			0.507
I	7 (5.00)	5 (3.70)	
II	86 (61.43)	92 (68.15)	
III	32 (22.86)	22 (16.30)	
Unknown	15 (10.71)	16 (11.85)	
Histological type			0.746
Ductal	132 (94.29)	125 (92.59)	
Others	8 (5.71)	10 (7.41)	
21-gene recurrence score			0.220
Low risk	4 (2.86)	4 (2.96)	
Median risk	65 (46.43)	79 (58.52)	
High risk	56 (40.00)	43 (31.85)	
Unknown	15 (10.71)	9 (6.67)	
Radiation therapy			0.106
No	80 (57.14)	63 (46.67)	
Yes	60 (42.86)	72 (53.33)	
Endocrine therapy			0.226
SERM-based	65 (46.43)	52 (38.52)	
AI-based	75 (53.57)	83 (61.48)	

**Table 2 cancers-14-03221-t002:** The neutropenia rates in both arms.

	EC *n* = 140 (%)	TC *n* = 135 (%)	Risk Difference (95% CI)
Grade 3–4 neutropenia	71 (50.71)	65(48.15)	−0.100, 0.151
All grade neutropenia	131 (93.57)	100 (74.07)	0.103, 0.287

**Table 3 cancers-14-03221-t003:** Other adverse hematological events.

	EC *n* = 140 (%)	TC *n* = 135 (%)	*p*
Anemia			0.0007
All grade	60 (42.86)	31 (22.96%)	
Grade 3–4	1 (0.71)	0 (0)	/
Thrombocytopenia			
All grade	10 (7.14)	6 (4.44)	0.4852
Grade 3–4	2 (1.43)	1(0.74)	/
Hepatotoxicity			
All grade	16 (11.43)	11 (8.15)	0.4769
Grade 3–4	1 (0.71)	1 (0.74)	/

**Table 4 cancers-14-03221-t004:** Non-hematological toxicity by group.

	EC *n* = 140 (%)		TC, *n* = 135 (%)	
	Any Grade	Grade 3–4	Any Grade	Grade 3–4
Nausea/vomiting	123 (87.86)	2 (1.43%)	75 (55.56)	0 (0.00)
Hair loss	126 (90.00)	/	103 (76.30)	/
Nail change	76 (54.29)	/	30 (22.22)	/
Fatigue	55 (39.29)	/	61 (45.19)	/
Ostalgia	27 (19.29)	0 (0.00)	27 (20.00)	0 (0.00)
Dizziness	11 (7.86)	0 (0.00)	10 (7.41)	0 (0.00)
Headache	11 (7.86)	0 (0.00)	3 (2.22)	0 (0.00)
Oral ulcer	9 (6.43)	1 (0.71)	8 (5.93)	0 (0.00)
Constipation	9 (6.43)	0 (0.00)	7 (5.19)	0 (0.00)
Diarrhea	5 (3.57)	0 (0.00)	7 (5.19)	0 (0.00)
Decreased appetite	5 (3.57)	0 (0.00)	7 (5.19)	0 (0.00)
Insomnia	4 (2.86)	0 (0.00)	4 (2.96)	0 (0.00)
Rash/eczema	3 (2.14)	0 (0.00)	6 (4.44)	0 (0.00)
Stomachache	2 (1.43)	0 (0.00)	3 (2.22)	0 (0.00)
Cough	2 (1.43)	0 (0.00)	6 (4.44)	0 (0.00)
Infection	2 (1.43)	0 (0.00)	1 (0.74)	0 (0.00)
Numbness of limbs	3 (2.14)	0 (0.00)	5 (3.70)	0 (0.00)

**Table 5 cancers-14-03221-t005:** Univariate and multivariate analysis results for potential predictors of grade 3 or 4 neutropenia.

	Variable	Univariate		Multivariate	
		OR (95% CI)	*p*	OR (95% CI)	*p*
Regimen	EC vs. TC	0.902 (0.562, 1.448)	0.671	/	/
Age	<60 vs. ≥60	0.695 (0.400, 1.208)	0.197	/	/
Baseline neutrophil count	<2 vs. ≥2	0.61 (0.196, 1.932)	0.405	/	/
Comorbidity	no vs. yes	0.655 (0.406, 1.056)	0.083	0.669 (0.413, 1.084)	0.102
BMI	<25 vs. ≥25	0.605 (0.335, 1.091)	0.095	0.651 (0.358, 1.183)	0.159
Surgery	Mastectomy vs. BCS	0.615 (0.404, 1.049)	0.078	0.667 (0.412, 1.081)	0.101

## Data Availability

The data in the study are available on request from the corresponding author.

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
