# Peer review of "The Comparative Safety of Epirubicin and Cyclophosphamide versus Docetaxel and Cyclophosphamide in Lymph Node-Negative, HR-Positive, HER2-Negative Breast Cancer (ELEGANT): A Randomized Trial"

_cancers, 2022, doi:10.3390/cancers14133221_

Round 1

Reviewer 1 Report

This study was reported the open-label, randomized trial in patients with breast cancer who received adjuvant chemotherapy. Overall, this paper is well written. The reviewer thinks that this report has useful information for readers. The reviewer would like to suggest some critiques as follows.

1.     On line 36, the primary endpoint in this study was the grade3/4 neutropenia. However, study design was a open-label, randomized, non-inferior study with bone marrow suppression. Which is true?

2.     The author should describe clearly about the definition of the experimental group and the controlled group (control group?) at the Trial design section.

3.     Based on the results of this study, which is the recommended treatment regimen as an adjuvant treatment for the patients with breast negative lymph node involvement, HR-positive and HER-2 negative breast cancer.

Author Response

Comment 1:

On line 36, the primary endpoint in this study was the grade 3/4 neutropenia. However, study design was an open label, randomized, non-inferior study with bone marrow suppression. Which is true?

Response:

Thank you for your comment! The primary endpoint of the trial is the incidence of grade 3/4 neutropenia. In the section of methods, we made some revisions to specify the trial design.

Comment 2:

The author should describe clearly about the definition of the experimental group and the controlled group (control group?) at the Trial design section.

Response:

Thank you for the comment! Patients were randomized into the experimental group (EC) or control group (TC) as stated in the last paragraph of Trial design section.

Comment 3:

Based on the results of this study, which is the recommended treatment regimen as an adjuvant treatment for the patients with breast negative lymph node involvement, HR-positive and HER-2 negative breast cancer.

Response:

According to the results of our study, EC is non-inferior to TC as for the incidence of grade 3 or 4 neutropenia. However, more nausea, vomiting, hair loss and nail change were reported in EC group. Taking both efficacy and safety into consideration, TC is preferred in patients with node-negative, HR positive and HER2-negative breast cancer. In the meantime, EC regimen is an optional alternative for these patients. According to this comment, recommendation is added in the conclusion section at the end of the manuscript.

Reviewer 2 Report

Reviewer comments:

Comments to the Author

This manuscript describes the adjuvant therapy involving epirubicin and cyclophosphamide (EC), docetaxel and cyclophosphamide (TC) are both optional chemotherapy regimens for lymph node negative, hormone receptor (HR)-positive, human epidermal receptor 2 (HER2)-negative breast cancer patients. Authors concerned about the neutropenia being one of the most common adverse events (AEs) of these regimens. According to this study, the rate of grade 3-4 neutropenia varies in different studies and safety profile directly comparing EC and TC are lacking. However, the experimental designing is very limited in terms of confirming the data, and the authors didn’t pay attention to write manuscript with substantial evidence of confirmatory and supplementary data. Based on the data provided, discussion is extrapolating their findings. Authors need to go through the guidelines of the Journal policies and guidelines.

Major criticisms

·        Authors mentioned that patients were informed, however, whether the consent from the patients were signed. Whether the study was granted by any institutional committee. Please provide the statement.

·        Tissue staining for histology and FISH assay protocols are missing.

·        Figure legends are missing. Authors need to clarify and explain in detail about the data presented including the number of patients in that data.

·        No comparison was provided to show difference between epirubicin and cyclophosphamide (EC); and docetaxel and cyclophosphamide (TC) groups.

·        Authors need to describe “Table 2. Neutropenia rate of both arms” in the result section. What are conclusive facts for showing this table.

·        Please undergo a thorough check of the manuscript for typographical and grammatical errors.

Author Response

Comments to the author:

This manuscript describes the adjuvant therapy involving epirubicin and cyclophosphamide (EC), docetaxel and cyclophosphamide (TC) are both optional chemotherapy regimens for lymph node negative, hormone receptor (HR)-positive, human epidermal receptor 2 (HER2)-negative breast cancer patients. Authors concerned about the neutropenia being one of the most common adverse events (AEs) of these regimens. According to this study, the rate of grade 3-4 neutropenia varies in different studies and safety profile directly comparing EC and TC are lacking. However, the experimental designing is very limited in terms of confirming the data, and the authors didn’t pay attention to write manuscript with substantial evidence of confirmatory and supplementary data. Based on the data provided, discussion is extrapolating their findings. Authors need to go through the guidelines of the Journal policies and guidelines.

Respond to the comments:

Thank you for your comment!

A previous study found that ethnic differences exist between Asian and Caucasian patients with a much higher grade ≥ 3 neutropenia rate in TC regimen in Asian patients (30.7% vs 4.0%, p < 0.001)[1]. Our study provided a full description of safety profiles, especially grade 3 or 4 neutropenia rate in adjuvant EC and TC regimens in patients treated in Ruijin Hospital, through these results, we could take peep at the tolerance and compliance of Asian patients with EC and TC chemotherapy.

ELEGANT is a single-center, open-label trial, there is no arguing that limitation and bias exist without multi-center inclusion and blinding. However, the statistical considerations were well designed. The fixed margin and assuming neutropenia rate in EC group, which are determinant factors of sample size, were determined after comprehensive investigation and estimates of the effect of comparator in previous studies [2,3,4]. Therefore, regarding the primary endpoint, we believe the trial and statistical design was valid to answer the question whether grade 3 or 4 neutropenia rate of EC is non-inferior to that of TC in patients treated in Ruijin Hospital.

Major criticisms

  1. Authors mentioned that patients were informed, however, whether the consent from the patients were signed. Whether the study was granted by any institutional committee. Please provide the statement.

Response:

As stated in the second paragraph of the Trial design section, all subjects provided written informed consents. The protocol, consent and relative documents were approved by Independent Ethics Committee in Ruijin Hospital (Number 2015 [55]). The consent form and institutional approval of the trial will be uploaded in supplementary materials.

  1. Tissue staining for histology and FISH assay protocols are missing.

Response:

IHC assessment of ER (SP1, DAKO), PgR (PgR 636, DAKO), Ki67 (MIB-1, DAKO) and HER2 (4B5, Roche) were made from 4-μm slices of paraffin-embedded tumor samples by Ventana autostain system, BenchMark XT, and evaluated with internal and positive controls. ER/PR status was defined as positive if there are at least 1% positive tumor nuclei in the sample according to the recommendation of the American Society of Clinical Oncology (ASCO) and College of American Pathologists (CAP) guidelines. HER2 status was considered as negative if it scored 0 to 1+ by IHC. Fluorescent in situ hybridization was performed to all IHC 2+ and 3+ to determine the HER2 gene amplification. HER2 expression either with IHC 3 + or FISH amplified (ratio of HER2 to CEP17 of ≥ 2.0 or with a mean HER2 copy number ≥ 6) was considered positive.

Pathological assessment was revised in Trial Design section in the updated manuscript.

  1. Figure legends are missing. Authors need to clarify and explain in detail about the data presented including the number of patients in that data. 

Response:

Figure legends and clarifications were added in the revised manuscript.

  1. No comparison was provided to show difference between epirubicin and cyclophosphamide (EC) and docetaxel and cyclophosphamide (TC) groups. 

Response:

The study was aimed at comparing the safety and efficacy endpoints of EC versus TC regimen with all endpoints elaborated in the Results section and summarized in the Discussion Section. We have made some revisions to highlight the main findings. According to the findings of the trial, EC was non-inferior to TC in terms of grade 3/4 neutropenia. All-grade neutropenia, anemia, thrombopenia were similar between two arms. Non-hematological AEs were generally similar between two arms except that nausea/vomiting, hair loss and nail change were more common in the EC group. As for the survival endpoint, no statistical difference was observed in DFS for two groups.

  1. Authors need to describe “Table 2. Neutropenia rate of both arms” in the result section. What are conclusive facts for showing this table.

Response:

As the primary endpoint of the trial, grade 3 and 4 neutropenia rate were showed in Table 2, 50.71% and 48.15% separately in both groups. According to the pre-defined statistically meaningful difference margin of 0.15, the study met its primary endpoint.

All grade neutropenia rate is also a concerned endpoint in our study, which of both groups were displayed in Table 2 to provide a better vision of myeloid suppression effect of EC and TC regimens.

An annotation has been added to the relevant part in manuscript to locate the conclusive statement concerning Table 2.

  1. Please undergo a thorough check of the manuscript for typographical and grammatical errors.

Response:

The revised manuscript has been checked and corrected for language mistakes. Further English modification will also be provided by editorial office.

Reference:

  1. Chow, L. W. C., et al. (2017). "Toxicity profile differences of adjuvant docetaxel/cyclophosphamide (TC) between Asian and Caucasian breast cancer patients." Asia Pac J Clin Oncol 13(6): 372-378.
  2. Pico C, Martin M, Jara C, Barnadas A, Pelegri A, Balil A, et al. Epirubicin-cyclophosphamide adjuvant chemotherapy plus tamoxifen administered concurrently versus sequentially: randomized phase III trial in postmenopausal node-positive breast cancer patients. A GEICAM 9401 study. Ann Oncol. 2004;15(1):79-87.
  3. Vici P, Brandi M, Giotta F, Foggi P, Schittulli F, Di Lauro L, et al. A multicenter phase III prospective randomized trial of high-dose epirubicin in combination with cyclophosphamide (EC) versus docetaxel followed by EC in node-positive breast cancer. GOIM (Gruppo Oncologico Italia Meridionale) 9902 study. Ann Oncol. 2012;23(5):1121-9.
  4. Jones RL, Walsh G, Ashley S, Chua S, Agarwal R, O'Brien M, et al. A randomised pilot Phase II study of doxorubicin and cyclophosphamide (AC) or epirubicin and cyclophosphamide (EC) given 2 weekly with pegfilgrastim (accelerated) vs 3 weekly (standard) for women with early breast cancer. Br J Cancer. 2009;100(2):305-10.

Round 2

Reviewer 2 Report

The manuscript looks better after revision to be accepted.